

# Preparation of hybrid calibrated absorption cross sections for a compact UV-DOAS measurement

Beni Adi Trisna[1,2], Sang Woo Kim[1,3], Yong-Doo Kim[4], Miyeon Park[1,2], Seung-Nam Park[5], Jeongsoon Lee[1,2]

[1]Greenhouse Gas Metrology team, Korea Research Institute of Standard and Science (KRISS), 267 Gajeong-ro, Yuseong-gu, Daejeon 34113, Republic of Korea
[2]Science of Measurement, University of Science and Technology (UST), 217 Gajeong-ro, Yuseong-gu, Daejeon 34113, Republic of Korea
[3]Sunkyungkwan University (SKKU), 2066 Seobu-ro, Jangan-gu, Suwon 16419, Republic of Korea
[4]Center for Gas Analysis, Korea Research Institute of Standard and Science (KRISS), 267 Gajeong-ro, Yuseong-gu, Daejeon 34113, Republic of Korea
[5]National Centre for Standard Reference Data (NCSRD), Korea Research Institute of Standard and Science (KRISS), Daejeon 34113, Republic of Korea

*Correspondence to*: Jeongsoon Lee (leejs@kriss.re.kr)

**Abstract.** A low-cost differential optical absorption spectroscopy (DOAS) spectrometer has been calibrated using a hybrid approach in which multiple absorption cross sections (ACSs) are measured and compared with previously obtained ACSs. The ACSs obtained from the measurement in this study are referred to as precise measurement

(PM) ACSs, while those obtained from the simulation are referred to as hybrid simulation (HS) ACSs. The compact UV grating spectrometer was used to measure PM ACSs at different pressures and temperatures ranging from 295.75 K to 298.15 K. The spectral range was 230-320 nm with a spectral resolution of 0.28-0.39 nm. Uncertainties were evaluated and traceable to SI units. The relative standard measurement uncertainty of the PM ACS for o-xylene, p-xylene, SO2, benzene, styrene and toluene were 3.1%, 3.1%, 3.1%, 3.0%, 3.0% and 2.9%.

respectively. This combined relative standard uncertainty includes contributions from optical path length (1.4%), CRM concentration (≤0.74%), absorbance repeatability (≤1.5%), pressure (0.78%) and temperature (2.1%). ACSs for reactive trace gases without CRMs were derived using the HS method. This involved convolving literature spectra with measured instrumental functions (IFs) and subsequently applying spectral fitting to achieve ACSs that are well-aligned with those that would be obtained if measured with this spectrometer. The uncertainty of the

HS ACSs results from the referenced ACSs, the determination of the IFs and the fitting procedures. The uncertainty of the referenced ACSs is taken directly from the literature and ranges from 0.05% for benzaldehyde and formaldehyde to 5% for oxygen. The uncertainty of the IFs is 0.67%, 0.74% and 0.65% in regions I, II and III respectively. The total uncertainty for HS ACS is estimated to be 0.25% for benzaldehyde, 1.01% for NO2, 0.95% for formaldehyde, 0.75% for p-cresol, 0.74% for m-xylene, 4.13% for phenol, 4.19% for ethylbenzene,

2.83% for ozone in region I, 3.51% for ozone in region II, 5.24% for oxygen in region I and 5.31% for oxygen in region II, taking into account the above uncertainty components. In a laboratory measurement of a gas mixture (benzene, toluene and ortho-xylene), the difference between the measurement values of the certified reference material (CRM) produced by the gravimetric method and the DOAS measurement is 8.5% for benzene, 1.6% for toluene and 4.9% for ortho-xylene. Therefore, the uncertainty of the DOAS system was estimated to be 15.2% for

benzene, 8.2% for toluene and 11.6% for o-xylene, taking into account the uncertainties of the ACS (≤3.1%), the



fitting procedure (≤0.6%), the difference with the CRM value (≤8.5%) and the path length (1.4%). In general, the system is ready for use in field measurements using the long-path DOAS technique.

**List of Abbrevations and Variables**

| Abbreviation | Definition |
| --- | --- |
| ACSs | Absorption cross sections |
| BIPM | International bureau of weights and measures |
| BTX | Benzene, toluene, and xylenes |
| CCD | Charge-coupled device |
| $CO_2$ | Carbon dioxide |
| CRM | Certified reference material |
| DL | Detection limit |
| DOAS | Differential optical absorption spectroscopy |
| DUV | Deep ultraviolet |
| FTIR | Fourier-transform infrared spectroscopy |
| FWHM | The full width at half maximum |
| Hg | Mercury |
| HS | Hybrid simulation |
| IF | Instrument function |
| ISO | The international organization for standardization |
| KRISS | Korea research institute of standards and science |
| LP-DOAS | Long-path differential optical absorption spectroscopy |
| $N_2$ | Nitrogen |
| $N_2O$ | Nitrous oxide |
| NLS | Nonlinear least squares |
| $NO_2$ | Nitrogen dioxide |
| $O_2$ | Oxygen |
| $O_3$ | Ozone |
| PAHs | Polycyclic aromatic hydrocarbons |
| PM | Precise measurement |
| PRT | Platinum resistance thermometer |
| SI | The international system of units |
| $SO_2$ | Sulfur dioxide |
| UV | Ultraviolet |
| VOCs | Volatile organic compounds |




| Variable | Definition |
| --- | --- |
| $A(\lambda)_{meas}$ | Absorption spectrum measured at wavelength $\lambda$ |
| $A(\lambda)_{HS}$ | Absorption spectrum simulated using HS approach at wavelength $\lambda$ |
| $A_{ref}(\lambda)$ | Reference absorption spectrum from literature |
| $\sigma_i(\lambda, p, T)$ | ACS of gas species $i$ at wavelength $\lambda$, pressure $p$, and temperature $T$ |
| $\varepsilon_R(\lambda)$ | Extinction from air molecules (Rayleigh scattering) |
| $\varepsilon_M(\lambda)$ | Extinction from aerosol particles (Mie scattering) |
| $I(\lambda, L)$ | Intensity of light at wavelength $\lambda$ after passing optical path $L$ |
| $I_0(\lambda, L)$ | Initial light intensity without gas species |
| $\sigma_{PM}(\lambda)$ | ACS from precise measurement (PM) technique |
| $\sigma_{HS}(\lambda)$. | ACS from HS technique |
| $a$ | Wavelength squeezing coefficient in HS |
| $b$ | Wavelength shifting coefficient in HS |
| $\kappa$ | Intensity scaling factor in HS |
| $c_j$ | Concentration of gas species related to the ACS from PM |
| $c_k$ | Concentration of gas species related to the ACS from HS |
| $R$ | Universal gas constant |
| $u_{c,r}(\sigma_{PM}(\lambda))$ | Combined relative standard uncertainty from PM |
| $u_{L,r}$ | Relative standard uncertainty due to path length determination |
| $u_{c_j,r}$ | Relative standard uncertainty of CRM from gravimetric certificate |
| $u_{UV,r}$ | Relative standard uncertainty of UV light stability |
| $u_{rep,r}$ | Relative standard uncertainty of repeatability of absorbance |
| $u_{p,r}$ | Relative standard uncertainty due to pressure measurement |
| $u_{T,r}$ | Relative standard uncertainty due to temperature measurement |
| $u_{c,r}(\sigma_{HS}(\lambda))$ | Combined relative standard uncertainty from HS |
| $u_{\sigma_r(\lambda)}$ | Relative standard uncertainty from the ACS literature |
| $u_{IF_{convol}(\lambda),r}$ | Uncertainty in the determination of instrument function |
| $u_{fit,r}$ | Uncertainty due to NLS fitting process |






## 1 Introduction

Toxic gases have been substantially released into the urban atmosphere since the rapid growth of industries and
mega-cities. These pollutants, including polycyclic aromatic hydrocarbons (PAHs) and volatile organic
compounds (VOCs) such as benzene, toluene, sulfur dioxide ($SO_2$), styrene, and xylenes, originate from both
indoor and outdoor environments. Major sources of these emissions include petrochemical industries, oil
refineries, motor vehicles, and power plant smokestacks (Johansson et al., 2014; Liu et al., 2016; Skorokhod et
al., 2017; Wei et al., 2017). Their active reactions in the atmosphere contribute to the formation of secondary
pollutants (Jenkin and Clemitshaw, 2000). Estimating emission inventories and understanding the photochemistry
of these compounds are crucial, requiring continuous monitoring of gaseous components such as VOCs, $SO_2$, and
$NO_2$. The use of compact monitoring devices enables the effective implementation of policies and actions to
manage emissions, thereby promoting the restoration of clean and safe air in a cost-effective manner.

Among numerous methods for measuring gaseous VOCs, SO2, and NO2, optical techniques such as
differential optical absorption spectroscopy (DOAS) (Platt and Stutz, 2008; Lee and Kim, 2003; Lee et al., 2002)
and Fourier-transform infrared spectroscopy (FTIR) (Fally et al., 2009) has been widely used in monitoring
stations. Absorption spectroscopy is particularly advantageous due to its ability to non-destructively quantify gas
concentrations over a defined optical path length in the atmosphere, with minimal maintenance requirements.
Beyond their economic benefits, small and lightweight spectrometers offer the additional advantage of easy
deployment for outdoor air quality monitoring. However, in terms of detector performance—such as sensitivity
and pixel spacing—these compact spectrometers typically have lower performance compared to high-resolution
spectrometers. Accordingly, studies of outdoor demonstrations using compact spectrometers are also widely
reported with the growing interest in performance evaluation.

To generate reliable air quality monitoring data, it is crucial to produce accurate ACSs. While the ideal
approach is to create ACSs using the same spectrometer intended for monitoring, an alternative method involves
convoluting published ACSs. When using the published ACS, it is recommended to select high resolution ACSs
to simulate ACSs by convoluting them with the instrument function (IF) of the spectrometer in use, as
demonstrated by Fally et al. (2009). Typically, this convolution assumes that the IF of the spectrometer remains
constant across the measurement wavelength range; however, in reality, the IF varies with wavelength. This
assumption can lead to significant discrepancies between measured and simulated ACSs. For instance, when
comparing $SO_2$ absorption spectra obtained by convoluting spectral databases at various resolutions—ranging
from low (0.16 to 0.49 nm) to high (FTIR: 16 cm$^{-1}$ and grating spectrometer: 0.05 to 0.11 nm)—with the IF, we
observed a substantial difference of up to 62% ($8.4 \times 10^{-4}$) in peak-to-peak optical density between the convoluted
and measured spectra.

Since Pantos et al. (1978) measured the absolute ACSs of benzene in the temperature range of 293.15 to
300.15 K within the 135 to 270 nm region at resolutions of 0.08 and 0.25 nm, subsequent studies have reported
ACSs in the 130–280 nm region at resolutions ranging from 0.065 to 0.2 nm (Hiraya and Shobatake, 1991; Suto
et al., 1992). Additionally, ACSs for monocyclic aromatic hydrocarbons have been measured in the 220–290 nm
region with resolutions between 0.11 and 0.2 nm (Milton et al., 1992; Trost et al., 1997; Etzkorn et al., 1999) and
in the UV region (238–270 nm) with improved spectral resolution of 1 cm$^{-1}$ (Fally et al., 2009). More recently,
Klingbeil et al. (2007) reported the temperature dependence of ACSs from 298.15 to 773.15 K in the mid-IR
region with a spectral resolution of 1 cm$^{-1}$ using FTIR. In addition to aromatic hydrocarbons, absolute ACSs of



SO₂ have been reported at temperatures of 295 and 210 K in the 300–325 nm region with a resolution of 0.03 nm (Mcgee and Burris Jr, 1987), and at 295 K in the 250–370.37 nm region with resolutions of 2 and 16 cm⁻¹ using

a Fourier-transform spectrometer (Vandaele et al., 1994). Although the accuracy of ACSs has improved over time, the uncertainties associated with these measurements have rarely been thoroughly assessed.

In this study, we aimed to generate accurate ACSs either through precise measurement (PM) or hybrid simulation (HS) and to estimate their uncertainties, which are crucial for the application of compact spectrometers in air quality monitoring. We present an approach for generating accurate hybrid ACSs to ensure reliable results

in DOAS observations. The hybrid simulation method combines convolution and spectral fitting to the actual spectral pixels on the instrument detector array. After applying our method, the discrepancy between the hybrid and measured ACSs was reduced by up to 8.8% (2.7×10⁻⁴) in the case of SO₂ absorption. The measurement uncertainties of ACSs have been thoroughly assessed, ensuring that the observations are traceable to the International System of Units (SI).

**2 Experimental details**

**2.1 Instrumental setup for ACS measurement**

Figure 1 provides a schematic overview of the ACS measurement setup. A xenon arc lamp (E7536, Hamamatsu, Shizuoka, Japan) was collimated using a lens with a focal length of 5 cm, allowing the beam to pass parallel through a gas cell that is 50 cm in length and 2.54 cm in diameter. ACS measurements were conducted by scanning

the CRM gas spectrum following the acquisition of a background spectrum. Each measurement was repeated five times to ensure accuracy and reproducibility.

First, the background spectrum was measured through an evacuated cell after the cell had been purged with N₂ gas at 10⁻¹ mbar or less. Second, the CRM spectrum was obtained from a gas cell filled with CRM to the target pressure after several flushes, maintained at a specified temperature and pressure. The cell pressure was

measured with a relative standard uncertainty of 0.4% using a pressure transducer (MKS 626C13TAE, MA, USA), which was calibrated against an ultrasonic interferometer manometer at KRISS. Temperature measurements were conducted using a thermohygrometer (Testek, 303A, OH, USA) equipped with a PRT sensor probe attached to the cell surface. The PRT sensor was calibrated against a standard platinum resistance thermometer (Fluke-Hart Scientific, 5628, WA, USA), resulting in a measurement uncertainty of approximately 1%. CRM cylinders were

allowed to equilibrate in the laboratory for 3 to 5 days prior to measurement.

A Czerny-Turner type mini spectrometer (Avantes AVASpecs-ULS2048LTEC-USB2, Apeldoorn, Netherlands), which is both lightweight and compact, was employed. To obtain signals from 237 nm to 355 nm, the spectrometer was equipped with a 2400 line/mm grating, an entrance slit measuring 50 μm wide × 1000 μm high, and a charge-coupled device (CCD; Sony 2048-pixel CCD linear image sensor, ILX511, TYO, Japan). A

DUV lumogen-coated CCD was used to prevent second-order UV response effects and enhance sensitivity, and it was cooled to 278.15 K to reduce thermal noise and signal distortion. The spectra were acquired using integrated software provided by the spectrometer manufacturer and were processed using a custom fitting program to quantify the gas concentration in the cell.





### 2.2 Preparation of CRMs

All standard mixtures were prepared gravimetrically in accordance with ISO 6142: Gas Analysis–Preparation of Calibration Gas Mixtures–Part 1: Gravimetric Method for Class I Mixtures (ISO, 2015) (Guide, 2016), by diluting high-purity raw materials with high-purity nitrogen, as detailed in a previous study (Kang et al., 2021). A 10 L electro-polished aluminum cylinder (Luxfer, Nottingham, UK) was used to minimize adsorption and potential reactions on the cylinder surface or between components of the mixture. Before filling, each cylinder was

evacuated using a turbo pump (Varian TV301-NAV, USA) to a pressure of $10^{-6}$ bar while being heated to approximately 340 K for two days to remove any residual moisture.

     To accurately determine the mass of gas, two highly precise mass balances (Mettler-Toledo XP-26003L and AT201, Greifensee, Switzerland) were calibrated against Class E2 calibration weight reference standards (Metrology, 2004). After evacuation, a mass corresponding to the target molar fraction was injected via syringe

into a line connected to the evacuated cylinder, vaporized with an $N_2$ diluent, and transferred to the cylinder. The masses of the syringes before and after liquid loading, as well as the $N_2$ filling of the cylinder, were measured. During the weighing process, the laboratory was maintained at a temperature of $293 \pm 1$ K and a humidity of $50 \pm 10\%$. A detailed description of the CRM gas production process is available in previous studies (Kang et al., 2021; Kim et al., 2018).

Table 1 lists the concentrations and associated uncertainties of the CRMs used in this study. Benzene, toluene, ortho-xylene, para-xylene, styrene, and sulfur dioxide were prepared in a nitrogen balance, along with a mixture containing benzene, toluene, and ortho-xylene. The molar fractions of these compounds and their relative standard uncertainties, ranging from 0.45% to 0.74%.

### 2.3 Hybrid calibration of spectrometer's ACSs

**2.3.1 Instrument functions**

To produce accurate ACSs, the instrument function (IF) of the spectrometer must be determined. The IF, which characterizes the spectral features, can be obtained by measuring the emission line of a calibration lamp, such as a mercury (Hg) lamp (model: Pen-Ray 11SC, Analytik-jena). Hybrid simulation (HS) of ACSs (denoted as $A(\lambda)_{HS}$) involves converting the reference absorption spectrum from the literature (denoted as $A(\lambda)_{ref}$) using the

IF to simulate how the absorption spectrum would appear if measured by the specific spectrometer. This process, known as hybrid simulation (HS), is represented by the mathematical expression below (Platt and Stutz, 2008)

$$|A(\lambda)_{meas} - A(\lambda)_{HS}| \rightarrow min, \tag{1}$$

where

$$A_{HS}(\lambda) = c \cdot (A_{ref}(\lambda) \otimes IF_\lambda)_{\lambda_{this\,work} = a \cdot \lambda_{Lit} + b}, \tag{2}$$

As a form of virtual spectrometer, the IF cannot perfectly replicate the spectrum dispersed by the monochromator onto the CCD array pixels, as shown in equation (2). To achieve ACSs that are well-matched to our spectrometer, the ACS spectrum is simulated independently for each wavelength region, using the IF specific to that region. Each regional IF captures the optical characteristics of the corresponding area, where the absorption lines of each gas are simulated. This approach accounts for the differing resolutions across the three regions. The

three regions were selected based on the significant variations in the BTX absorption peaks.





As shown in Figure 2, the full width at half maximum (FWHM) of the IFs measured in different regions varied significantly, a phenomenon commonly observed in compact grating-type spectrometers with short focal lengths (Palmer and Loewen, 2005). To address this, the measured spectra were processed by dividing the fitting region into smaller, arbitrary slices to minimize residuals. Given that the nonuniformity of the FWHM substantially impacts the accuracy of gas concentration measurements, three distinct IFs were utilized in this study to mitigate this issue. Specifically, region I (238-249 nm) used the IF shown in Figure 2(a), region II (249-283 nm) used the IF depicted in Figure 2(b), and region III (283-305 nm) employed the IF illustrated in Figure 2(c). Each IF corresponds to a specific wavelength range.

**2.3.2 Hybrid simulation (HS) of the ACS**

The ACSs for oxygen, ozone, ethylbenzene, m-xylene, phenol, para-cresol, formaldehyde, and benzaldehyde were generated using HS method, with the with the categorization of the simulation range is summarized in Table 2. The reference spectra (Trost et al., 1997; Etzkorn et al., 1999; Meller and Moortgat, 2000; Voigt et al., 2001; Volkamer, 1996; Volkamer, 2001) were convoluted with the IFs determined in Section 2.3.1 using a custom processing tool developed in MATLAB software. The wavelength range was divided into three distinct regions, each with its own IF. Although absorptions are distributed across all three regions and multiple species coexist within each region, species with prominent absorptions were grouped and fitted together. Notably, oxygen and ozone exhibit strong absorption across all regions. Accordingly, three separate ACSs for ozone were independently created for each of the three regions, as detailed in Table 2.

The technical details of the reference ACSs, including cell length, concentration, wavelength range, and measurement technique, and resolution are summarized in Table 3. The ACSs reported by B. Trost et al. (1997) and Etzkorn et al. (1999) are comparable to those presented in this study. Both studies utilized spectrometers with spectral resolutions of 0.146 nm and 0.11 nm (FWHM), respectively, and measured the ACSs for ethylbenzene, benzaldehyde, meta-xylene, and para-cresol using samples prepared by introducing liquids into a 19 mm diameter quartz cell. All ACSs listed in the table were measured with spectral resolutions ranging from 0.025 to 0.2 nm.

**3 Results and discussions**

**3.1 Hybrid calibration of DOAS spectrometer**

From the Lambert-Beer's law, the intensity $I(\lambda,L)$ at the end of optical light path length $L$ from a light intensity $I_0(\lambda, L)$ in the presence of gas species can be derived as

$$I(\lambda, L) = I_0(\lambda, L) exp[-L(\sum \sigma_i(\lambda, p, T) \cdot c_i(l) + \varepsilon_R(\lambda, l) + \varepsilon_M(\lambda, l))], \tag{3}$$

where $\varepsilon_R(\lambda)$ is the extinction from air molecules (Rayleigh scattering) and $\varepsilon_M(\lambda)$ is the extinction from aerosols particles (Mie scattering). The spectral absorption of gas species $i$ is designated by its ACS $\sigma_i(\lambda, p, T)$, which depends on the wavelength $\lambda$, pressure $p$, and temperature $T$, and by its position along the optical path length $l$. The total absorption in the path must to be obtained from both precise measurement ($\sigma_{PM}(\lambda)$) and hybrid simulation ($\sigma_{HS}(\lambda)$). As a result, equation (3) can be modified to incorporate a hybrid combination of both measurement and simulation method, as follows:



$$I(\lambda, L) = I_0(\lambda, L)exp\left[-L\left(\sum \sigma_{PM}(\lambda) \cdot c_i + \sigma_{HS}(\lambda) \cdot c_j + \varepsilon_R(\lambda) + \varepsilon_M(\lambda)\right)\right], \quad (4)$$

For the $\sigma_{HS}(\lambda)$, we have compared the simulated ACS with the measured in the work for some species, such as benzene, para-xylene, and $SO_2$. As shown in Figure 5, the ACS obtained from the HS method is not

significantly different from the PM value because it is determined through the optimization of $a$, $b$, and $\kappa$ coefficients through solving NLS (Eq. 5) to equally fit the measured ACS.

$$\sigma_{HS}(\lambda) = \kappa \cdot (\sigma_{ref}(\lambda) \otimes IF_\lambda)_{\lambda_{this\,work}=a\cdot\lambda_{Lit}+b}, \quad (5)$$

where $a$, $b$, and $\kappa$ are the NLS fit coefficients of wavelength squeezing, wavelength shifting, and intensity scaling for the convoluted ACS. We then applied the NLS fit coefficients for a set of gas species at regions (Region I to

III) to let them match to the measured through our spectrometer. Finally, we obtained the mathematical expression for a hybrid calibration of DOAS spectrometer.

$$I'(\lambda, L) = I_0'(\lambda, L)exp\left[-L\left(\sum_{j=1}^{j=n} \sigma_{PM}'(\lambda) \cdot c_j + \sum_{k=1}^{k=n} \sigma_{HS}'(\lambda) \cdot c_k\right)\right], \quad (6)$$

where the prime symbol (′) indicates the differential quantities for the respective parameters, which is called hybrid calibration method to produce ACSs for this work.

**3.2 Determination of the precise measured ACSs, $\sigma_{PM}(\lambda)$**

Figure 3 presents the raw spectra recorded by the spectrometer for both the background and sample measurements for toluene gas as a representative example. To ensure consistent illumination, we aligned and checked the uniformity of the xenon lamp beam and verified its uniformity at both the entry and exit points of the fiber optic cable. The background spectrum (black line) displays a lamp profile with a sharp peak around 291 nm. However,

by dividing the sample spectrum by the background spectrum, this approach effectively eliminates the lamp profile and spectral noise, enhancing the quality of the ACS measurements and mitigating stray light and internal variations from the xenon lamp.

The ACSs of benzene, toluene, styrene, sulfur dioxide, ortho-xylene, and para-xylene measured in this study are presented in Figure 4. Table 4 provides a comprehensive summary of all ACS data, including those

measured in this work and those reported by others, along with the corresponding measurement parameters, such as resolution, temperature, pressure, and the chemicals used. It is important to note that the spectrometer employed in this study inherently sacrifices some performance quality in exchange for its compact size. As previously discussed, the shape of the peaks is influenced by both optical and environmental factors, including optical resolution and the temperature and pressure within the gas cell, as detailed in Table 4. The intensities of the

measured ACSs differed by up to 47.6% from the ACSs derived by convolving the literature data with the instrument function (IF) previously obtained in Figure 2 (a-c). The apparent ACSs from these measurements can be expressed as:

$$\sigma_{PM}(\lambda) = \frac{-\ln(A(\lambda)_{meas}) \cdot R \cdot T}{c_j pL}\bigg|_{\lambda=\lambda_0}, \quad (7)$$



In this context, $A(\lambda)$ represents the ratio of the measured transmittances with and without the gas species present

in the cell $\frac{I(\lambda)}{I_0(\lambda)}$, $L$ is the optical path length, $c_j$ is the mole fraction of the gas species, $p$ is the total pressure in the

cell, $T$ is the temperature, and $R$ is the universal gas constant. Although absorption due to electronic transitions is

generally less sensitive to environmental parameters such as temperature compared to vibrational transitions

(Grosch et al., 2010), the peak magnitudes in the ACS of Eq. (7) have been reported to be temperature-dependent.

This effect has been observed not only in the ultraviolet but also in the infrared spectra, where higher temperatures

result in sharper and more pronounced peaks in the ACS of hydrocarbons (Klingbeil et al., 2007; Fally et al.,

2009). These findings likely contribute to the differences observed between the ACS measurements in our study

and those reported in previous research.

Unlike the earlier studies (Fally et al., 2009; Trost et al., 1997; Etzkorn et al., 1999), which primarily

used methods involving the degassing of liquids into gas tubes, our study employed gaseous certified reference

materials (CRMs) under controlled conditions. The degassing of liquids into gas tubes has the disadvantage of a

tendency for the resulting gases to easily condense and adsorb onto the walls of the apparatus, making it difficult

to accurately quantify their concentration.

We evaluated the uncertainty in the measured ACS, considering factors as shown in Eq. (7). The total

combined relative standard uncertainties $u_{c,r}$ of the ACS were calculated using the general law of uncertainty

propagation (BIPM, 2008), resulting in the following:

$$u_{c,r}^2\big(\sigma_{PM}(\lambda)\big) = u_{L,r}^2 + u_{c_j,r}^2 + u_{UV,r}^2 + u_{rep,r}^2 + u_{p,r}^2 + u_{T,r}^2, \qquad (8)$$

The relative standard uncertainty because of light path length $u_{L,r}$ contributed to 1.4% of the relative uncertainty.

The relative standard uncertainty of CRM, $u_{c_j,r}$ obtained using the gravimetric technique vary depending on each

gas species, and the values are summarized in Table 1. The relative standard uncertainty because of absorbance

has two sources: the stability of UV light source $u_{UV,r}$ and the repeatability of absorbance $u_{rep,r}$. $u_{UV,r}$ was

estimated from the intensity stability of $I_0$ measurements for a day and the absorbance and $u_{rep,r}$ was calculated

by considering the standard deviation of four ACS measurements. Upon combining these two uncertainties, the

absorbance uncertainty values range between 1.0% (toluene) and 1.5% (o-xylene). The relative standard

uncertainty because of pressure $u_{p,r}$ and temperature $u_{T,r}$ measurements contributed 0.78% and 2.1%,

respectively. The result shows that $u_{c,r}$ is 3.0% (benzene and styrene), 3.1% (o-xylene, p-xylene and SO$_2$), and

2.9% for toluene. The main uncertainty component in the ACS calculation in this work is temperature, which

contributes approximately 47-54% to the total uncertainty.

The detection limits presented in Table 4 were calculated based on a 1000 m light path length. The

detection limit values (ppb·m·atm) were derived by applying the signal-to-noise ratio (Shrivastava and Gupta,

2011) to the following equation: $DL = \frac{3*Noise}{OD'_{max}}c \cdot p \cdot L$. (Etzkorn et al., 1999) also noted that the minimum

detectable optical density depends on the absorber concentration, visibility, and the field setup of LP-DOAS. In

this study, the detection limits for each gas species ranged from 0.01 ppb (para-xylene) to 0.54 ppb (ortho-xylene),

which is highly competitive given the low-resolution dispersion of the compact spectrometer.





### 3.3 ACS from hybrid simulation $\sigma_{HS}(\lambda)$

Figure 5 presents the ACSs from reference sources (blue line in the upper panel) and compares the ACS spectra obtained using the PM method (red line) with those from the HS method (black line) in the lower panel. The simulated ACSs were generated using parameters derived from the method outlined in Section 2.3. The HS procedure involved shifting, squeezing, linear multiplication, and convolution of the ACS spectra. The parameters applied in the non-linear least-squares (NLS) fit are summarized in Table 5.

When comparing the PM results to those processed with the HS method for various gas species, we found a strong agreement in the shape, position, and value of the ACS peaks, as shown in Figure 5 and summarized in Table 6. Notably, the discrepancy of up to 46% observed between the benzene ACS reported by Etzkorn et al. (1999) and this work was reduced to 0.8% after applying the HS method. The uncertainty in preparing ACSs using the HS method was found to be no more than 5.31%.

First-order fitting was performed by introducing shift and squeeze parameters to account for the non-uniformity of pixel spacing as a function of wavelength, resulting in a shift of the benzene peak position by 1 to 4 pixels (0.06 to 0.24 nm). Ultimately, applying the HS method led to an improved ACS, with peak positions matching within 1 pixel (0.06 nm).

Based on the successful application of the HS method, as shown in Table 6, the parameters derived for
benzene, para-xylene, and $SO_2$ (Table 5) were used to obtain the ACS for other species that could not be measured in the laboratory. The results for these additional gas species are presented in Figure 6, with details on the instrument function, peak location, and detection limits summarized in Table 7.

The detection limits for the gases measured in Table 7 were calculated for a path of 1,000 meters and were obtained by applying the same calculation method as for PM. From the simulated ACS spectra, the detection
limits of gas species range from 0.04 ppb (ozone in region I) to 0.80 ppb (nitrogen dioxide), which were comparable to references of the Table 7.

The ACS intensity is valid only within a range close to the instrument function used in the convolution process. As demonstrated in Figure 6 for O2 and O3, the ACS intensity values vary in the simulation results due to their broad-band absorption characteristics and the use of three different instrument functions. Therefore, when
conducting measurements with LP-DOAS, it is essential to adjust the fitting region for the gas to align with the ACS obtained from the HS process, ensuring it closely matches the measured IF.

Eq. 9 is used to evaluate the uncertainty of the ACS preparation process using the HS method. The uncertainty of HS ACS comes from the ACS literature used $u_{\sigma_r(\lambda)}$, the uncertainty in determining IF $u_{IF_{convol}(\lambda),r}$, and the uncertainty due to the non-linear fitting process $u_{fit,r}$. The uncertainty from the literature is taken from
each literature as shown in Table 7, although in some cases, such as $O_2$, the uncertainty value is not reported in detail, however, in the literature mentioned a pessimistic uncertainty value of 5%. Uncertainty due to IF is obtained from the standard deviation value of repeated measurements when making IF measurements repeated nine times. The result of this measurement is that the standard relative uncertainty in determining IF is 0.32%. Then this uncertainty value is propagated into the convolution and non-linear least squares fitting processes, so that each
uncertainty value due to IF is calculated for each gas species. Table 8 summarizes the uncertainty values for each component in the HS method. Finally, the uncertainty due to fitting error is taken from the standard deviation of the residual spectra in the NLS fitting process from region I to region III when determining the $a$, $b$, $\kappa$ coefficients. The same method of determining fitting error was also used by Trisna et al (2023) and Yi Hongming et al (2018)



when performing NLS fitting for $N_2O$ and $CO_2$ gases, respectively. Finally, after calculation, the total uncertainty

of HS ACS ranges from 0.25% (benzaldehyde) to 5.31% (oxygen 2).

$$u_{c,r}^2\big(\sigma_{HS}(\lambda)\big) = u_{\sigma_r(\lambda)}^2 + u_{IF_{convol},r}^2 + u_{fit,r}^2, \qquad\qquad (9)$$

### 3.4 Measurement of a gas mixture in the laboratory

To validate the PM ACSs, we measured the spectra of a VOC CRM mixture (cylinder D068005 in Table 1) in our

laboratory and then fitted the PM ACSs to the measured spectrum. The PM ACSs used to build the database are

the same as those shown in Figure 4. Figure 7 presents the results of evaluating the gas mixtures in the lab. A

differential optical density ($-\sigma'(\lambda)cL$) was obtained after applying high-pass filtering, logarithmic transformation,

and low-pass filtering to the $\frac{I(\lambda)}{I_0(\lambda)}$. We then fit the ACS to the mixture spectra using non-linear least-squares (NLS)

fitting, applying the Levenberg and Marquardt method along with the steepest descent method (Press et al., 1988).

The ACS of benzene, toluene, and o-xylene were then fitted subsequently with the mixture spectrum in the

wavelength interval between 240 and 280 nm in region II.

Table 9 summarizes the values obtained from the DOAS measurement alongside the corresponding CRM

values and their standard uncertainties. The differences between the two range from 1.6% for toluene to 8.5% for

benzene. The uncertainty in determining the concentration ($c$) in the optical density equation ($-\sigma'(\lambda)cL$) arises

from three factors: the uncertainty of the PM ACS $u_{c,r}(\sigma_{PM}(\lambda))$, the path length $u_{L,r}$, and the fitting method $u_{fit,r}$.

Due to the limitations of the fitting process, minimizing the cost function still leaves a residual value (ideally

zero). Therefore, following the general law of uncertainty propagation (BIPM, 2008), the relative total standard

uncertainty can be expressed by the following equation:

$$u_{c,r}^2(fv) = u_{c,r}^2\big(\sigma_{PM}(\lambda)\big) + u_{fit,r}^2 + u_{L,r}^2, \qquad\qquad (10)$$

The uncertainty from the PM ACSs was calculated using Eq. (8) from Section 3.2. The ACS uncertainty

contributes significantly to the DOAS measurements, making up 92.2% for o-xylene and 95.1% for both benzene

and toluene. This shows that accurately measuring the ACS is crucial for reducing overall uncertainty. The path

length contributes about 4.7% to the uncertainty for o-xylene and 4.9% for benzene and toluene. The fitting

uncertainty is minimal, being almost negligible for benzene and toluene, and about 3.1% for o-xylene. This fitting

uncertainty was determined by comparing optical density and peak-to-peak residual values (Platt and Stutz, 2008).

Table 9 shows that DOAS measurements, with an expanded uncertainty of 6-7%, deviate from CRM

values by 1.6-8.5%, with the largest difference being 8.5% for benzene. These differences are likely due to

adsorption or reactions on surfaces inside the gas cell and connecting lines. Specifically, benzene's sticky nature

(Mitchell et al., 2017; Mori, 1987) can cause large discrepancies between DOAS measurements and CRMs.

Additionally, before the actual DOAS measurement, we usually repeat the sample injection, vacuuming, and

nitrogen flushing two to three times. This process can naturally passivate the cell surface and connections, which

may lead to more gas being present than indicated by the pressure, resulting in DOAS values that are higher than

CRM values.





## 4 Conclusion

In this study, we successfully applied a hybrid calibration method for a long-path DOAS (LP-DOAS) system used
in atmospheric gas observation. This hybrid calibration involved two methods: direct precise measurement of
ACSs, known as the PM method, and simulation of ACSs using a virtual instrument based on the IFs of the
spectrometer, followed by a NLS fit, referred to as the HS method.

We successfully measured ACSs for benzene, toluene, styrene, sulfur dioxide, ortho-xylene, and para-
xylene. The standard uncertainties for these measurements were 3.1% for ortho-xylene, para-xylene, and sulfur
dioxide, 3.0% for benzene and styrene, and 2.9% for toluene. The main sources of uncertainty in the ACS
measurements were the light path length (1.4%), the concentration of calibration reference materials (≤0.74%),
absorbance determination (≤1.5%), pressure (0.78%), and temperature (2.1%). For ACSs obtained via the HS
method, the uncertainty was ≤5.31%, with the accuracy improved by dividing the convolution area into three
separate parts.

To validate the reliability of the prepared ACSs, we measured a gas mixture of benzene, toluene, and
ortho-xylene, prepared gravimetrically, using the spectrometer. The results, obtained through spectrum fitting,
showed discrepancies between DOAS retrieval and CRM values of 8.5% for benzene, 1.6% for toluene, and 4.9%
for ortho-xylene. The uncertainty components for the DOAS retrieval included the determination of ACSs
(≤3.1%), the fitting procedure (≤0.6%), and the path length (1.4%). The determination of ACSs significantly
influenced the accuracy of DOAS measurements, contributing 92.2% to the uncertainty for ortho-xylene and
95.1% for benzene and toluene. This underscores the importance of accurate ACS measurement for reducing
overall uncertainty in the DOAS measurement.

Controlled laboratory tests confirmed that the measured gas concentrations using the prepared ACSs
were accurate within 15%. These tests also showed that the measurement values can be linked to SI units, ensuring
that data collected during measurement campaigns are reliable and trustworthy.

## Acknowledgements

This work was funded by the Korea Meteorological Administration Research and Development Program under
Grant (KMI2022-01511). The funding agencies had no role in study design; in the collection, analysis, and
interpretation of data; in the writing of the report; and in the decision to submit the paper for publication.

## Data availability

Data are available at Zenodo at 10.5281/zenodo.13677342 (Beni, 2024).

## Conflict of interests

The authors declare that they have no known competing financial interests or personal relationships that could
have appeared to influence the work reported in this article.



**Author contributions**

BAT performed measurement, analyzed the data, and wrote the manuscript;

SK and MP helped BAT during CRM preparation;

YDK prepared gas CRM for ACSs measurement;

SP provided technical and managerial supports;

JL wrote the manuscript, conceived, and supervise the experiment;

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





# Figures and Tables


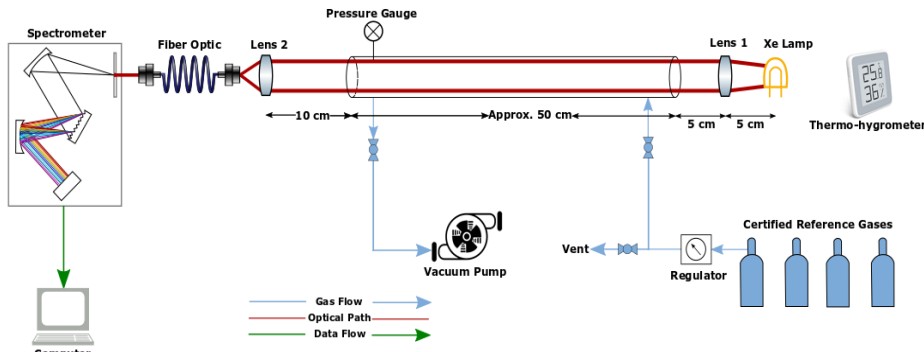

**Figure 1: Experimental setup of the ACSs measurements in the laboratory. A quartz cylinder tube of 50 cm in length was used to obtain the ACS.**









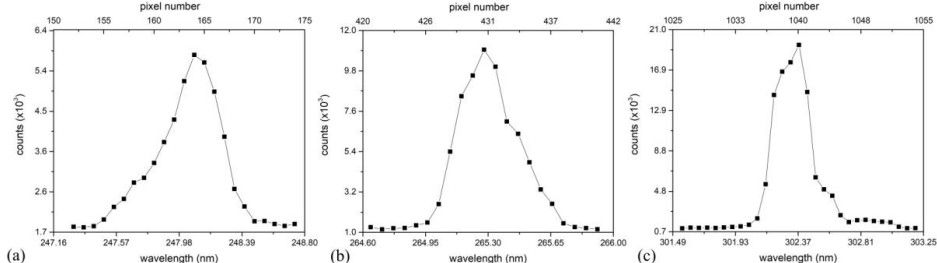

**Figure 2: Instrument function of a dispersive spectrometer at three wavelength regions obtained from Hg lamp (model: Pen-Ray 11SC, Analytik-jena ): (a) region I: at the center wavelength $x_c$ = 248.08 nm (FWHM = 0.39 nm); (b) region II: at the center wavelength $x_c$ = 265.28 nm (FWHM = 0.35 nm); and (c) region III: at the center wavelength $x_c$ = 302.33 nm (FWHM = 0.28 nm).**










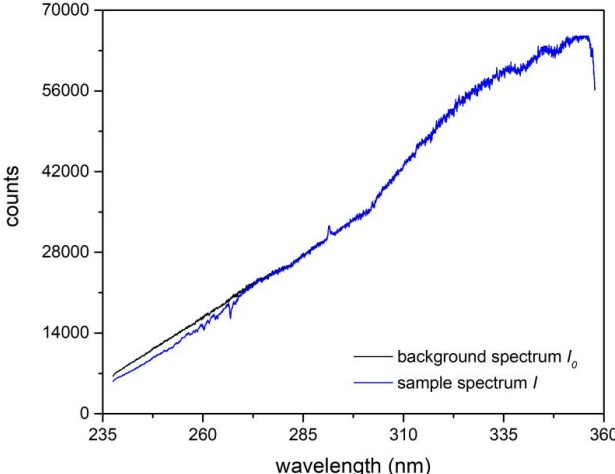

**Figure 3: Raw measured spectra for toluene, a representative gas. The black line shows the background spectrum $I_0$, and the blue line represents the sample spectrum $I$. The sample spectrum was taken at an ambient temperature of 545 295.8 K and a pressure of 224.86 Torr.**






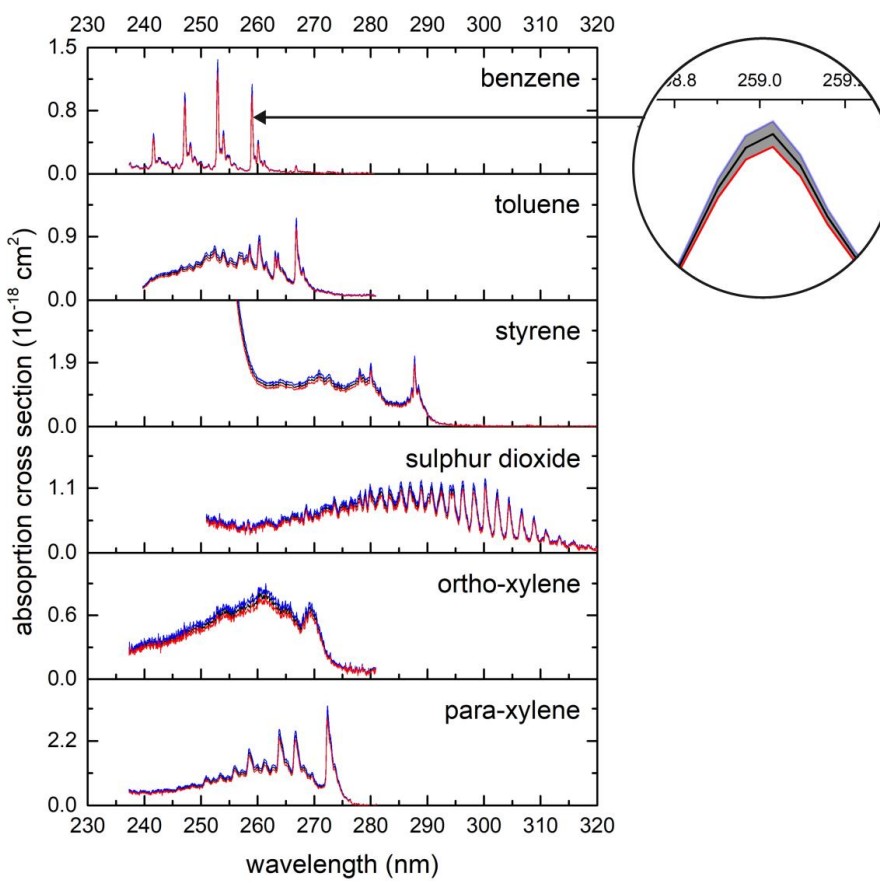

**Figure 4: The apparent UV absorption cross sections (ACSs) of benzene, toluene, styrene, sulfur dioxide, ortho-xylene, and para-xylene, measured in the laboratory, are shown by the black line. The relative expanded uncertainties of the measurements are indicated by the upper and lower bounds, represented by the blue and red lines, respectively. These uncertainties are as follows: 6.2% for ortho-xylene and sulfur dioxide, 6.0% for benzene, 5.8% for toluene, 6.0% for styrene, and 6.2% for para-xylene, at a 95% confidence level with a coverage factor $k = 2$.**



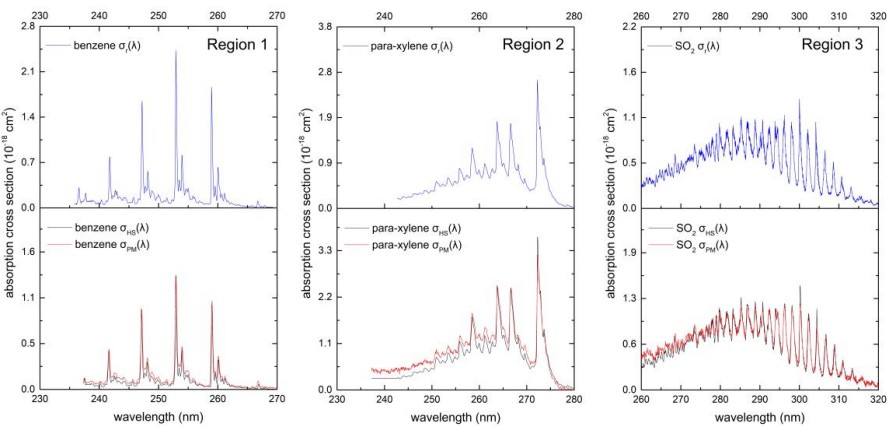

**Figure 5: The comparison between ACSs obtained using the precise measurement (PM) method and the hybrid simulation (HS) method is shown. The reference ACSs $\sigma_r(\lambda)$ are represented by the blue lines and were sourced from various literatures: benzene and para-xylene (T. Etzkorn et al., 1999) and SO₂ (Vandaele et al., 1994). After convoluting these reference spectra with the respective IFs for region I (benzene), region II (para-xylene), and region III (SO₂), a nonlinear least-squares (NLS) fit was applied to match the dispersion of pixels in the precise lab-measured ACSs $\sigma_{PM}(\lambda)$, represented by the red lines. This process results in the final hybrid simulation ACSs $\sigma_{HS}(\lambda)$, depicted by the black lines. The multiplicative factor from the least-squares fit addresses discrepancies arising from differences in spectrometer resolution, aligning the ACS values from the literature more closely with the measured ACSs.**




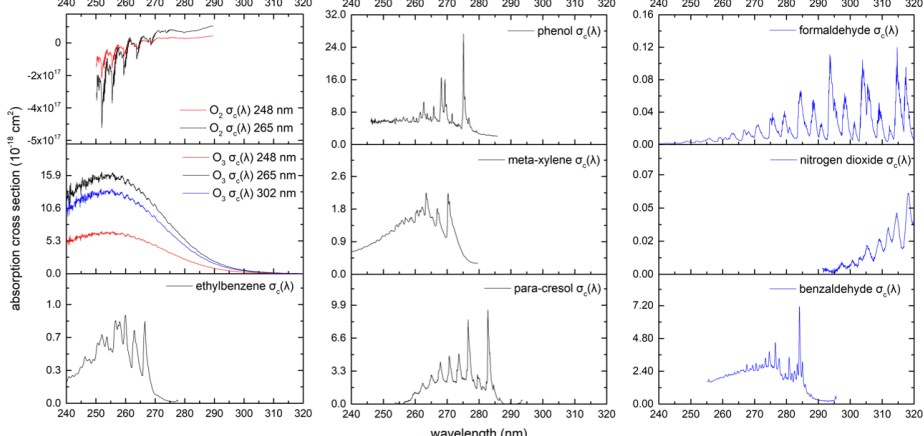


**Figure 6: Absolute UV absorption cross sections (ACSs) of oxygen, ozone, ethylbenzene, phenol, meta-xylene, para-cresol, formaldehyde, nitrogen dioxide, and benzaldehyde obtained from HS method. The colored line of the graphs indicates each instrument function used for convolution: red (248 nm), black (265 nm), and blue (302 nm).**





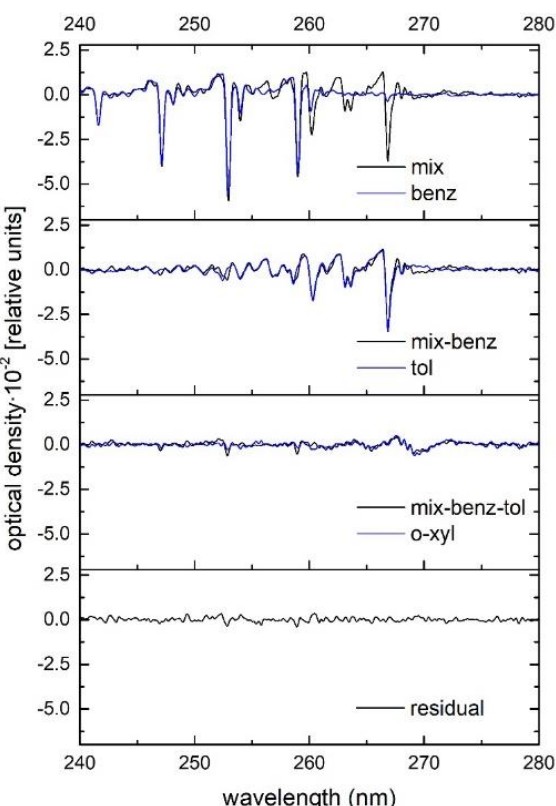

**Figure 7:** **This example demonstrates the evaluation of spectra measured in the lab. Each gas spectrum was sequentially subtracted from the mixture, with the black line representing the mixture spectrum and the blue line showing the fitting result multiplied by the ACS. The remaining residual spectra are displayed at the bottom, with an average residual value of $1.2 \times 10^{-4}$ (peak-to-peak).**





**Table 1 Summary of gas CRMs used in this study.**

| Cylinder code | Species | Amount of mole fraction (μmol/mol) | Standard uncertainty (Relative; %) |
|---|---|---|---|
| D641527 | Benzene | 503.16 | 0.45 |
| D746885 | Toluene | 487.59 | 0.45 |
| D581232 | Ortho-Xylene | 107.73 | 0.5 |
| D581015 | Para-Xylene | 100.46 | 0.7 |
| D769694 | Styrene | 94.847 | 0.65 |
| D500829 | Sulfur dioxide | 100.078 | 0.74 |
| D068005 | Benzene | 102.6 | 0.5 |
|  | Toluene | 101.0 | 0.5 |
|  | Ortho-Xylene | 100.8 | 0.5 |








**Table 2 Summary of target gas species for HS of ACSs considering different IFs in three regions.**

| Region | Wavelength [nm] | Gas species | Center wavelength of instrument function [nm] | Spectral resolution[a] [nm] |
|---|---|---|---|---|
| Region I (IF$_I$) | 238–249 | Oxygen, ozone | 248.08 | 0.39 |
| Region II (IF$_{II}$) | 249–283 | Oxygen, ozone, ethylbenzene, phenol, meta-xylene, para-cresol | 265.28 | 0.35 |
| Region III (IF$_{III}$) | 283–305 | Ozone, formaldehyde, nitrogen dioxide, benzaldehyde | 302.33 | 0.28 |

[a]Obtained from full width half maximum (FWHM) of mercury lines measurement






**Table 3 Technical data from ACS referenced for simulation.**

| Compound | Literature | Cell Length (cm) | Concentration (cmol/mol) | Wavelength (nm) | Technique[a] | Resolution (nm) |
|---|---|---|---|---|---|---|
| Oxygen | Volkamer et al. (Volkamer, 1996; Volkamer et al., 1998) | 6970 | 20.6 | 240–290 | X-G-PD | 0.05 |
| Ozone | Burrows et al. (Voigt et al., 2001; Burrows et al., 1999) | 2000 | 0.1–1 | 230–850 | X-FT | 0.027 at 230 nm |
| Ethylbenzene | Etzkorn et al. (Etzkorn et al., 1999) | 591 | 99 | 235–280 | D-G-PD | 0.146 |
| Phenol | Volkamer et al. (Volkamer, 2001) | 13,000 | 0.032 ppm | 245–285 | X-G-C | 0.2 |
| Meta-xylene | Trost et al. (Trost et al., 1997) | 2 | 99 | 230–290 | D-G-PD | 0.11 |
| Para-cresol | Trost et al. (Trost et al., 1997) | 2 | 99 | 230–290 | D-G-PD | 0.11 |
| Formaldehyde | Meller et al. (Meller and Moortgat, 2000) | 63 | >99.5 | 225–375 | D/T-G-PD | 0.025 |
| Nitrogen dioxide | Stutz et al. (Stutz et al., 2000) | 225 | 0.025–0.36 ppm | 290–405 | G-PD | 0.08 |
| Benzaldehyde | Etzkorn et al. (Etzkorn et al., 1999) | 591 | >99 | 250–300 | D-G-PD | 0.146 |

[a]D = deuterium lamp, X = xenon arc lamp, G = grating, FT = Fourier-transform spectrometer, C:CCD detector, PD = photodiode array, T: tungsten/halogen lamp



**Table 4 Comparison of the ACSs and detection limits obtained in this work with data from the literature.**

| Species | Literature | ACS of the strongest band (wavelength) [$10^{-18}$ cm$^2$ (nm)] | Standard uncertainty ($\sigma$) of the ACS [$10^{-18}$ cm$^2$] | Temperature (Pressure) [K (spec. pressure)] | Detection limit [ppb] | Assumed light path for detection limit calc. [m] |
|---|---|---|---|---|---|---|
| Benzene | This work | 1.31 (252.96) | 0.04 | 298.2 ± 1.0 (227.45 Torr)[b] | 0.05 | 1,000 |
| | Etzkorn et al. (1999) | 2.43 (252.94) | 0.21 | 298.15 ± 3.0 (vapor pressure)[c] | 0.39 | 480 |
| | Trost et al. (1997) | 2.5 (253.0) | 0.8 | 293.65 ± 1.0 (vapor pressure)[d] | $4.6 \times 10^{-5}$ | 2,000 |
| Toluene | This work | 1.07 (266.86) | 0.03 | 295.8 ± 1.0 (224.86 Torr)[b] | 0.50 | 1,000 |
| | Etzkorn et al. (1999) | 1.53 (266.64) | 0.24 | 298.15 ± 3.0 (vapor pressure)[c] | 0.66 | 480 |
| | Trost et al. (1997) | 1.4 (267.0) | 0.1 | 296.95 ± 0.1 (vapor pressure)[d] | $1 \times 10^{-4}$ | 2,000 |
| Styrene | This work | 1.97 (287.75) | 0.06 | 296.7 ± 1.0 (373.92 Torr)[b] | 0.03 | 1,000 |
| | Etzkorn et al. (1999) | 1.55 (287.7)[a] | 0.17 | 298.15 ± 3.0 (vapor pressure)[c] | 0.55 | 480 |
| Sulfur dioxide | This work | 1.19 (300.21) | 0.04 | 297.8 ± 1.0 (226.26 Torr)[b] | 0.12 | 1,000 |
| | Vandaele et al. (1994) | 1.03 (300.21) | 0.03 | 294.55 ± 0.4 (ATM)[e] | - | - |
| Ortho-xylene | This work | 0.70 (269.06) | 0.02 | 297.1 ± 1.0 (377.00 Torr)[b] | 0.54 | 1,000 |
| | Etzkorn et al. (1999) | 0.21 (269.00)[a] | 0.05 | 298.15 ± 3.0 (vapor pressure)[c] | 3.99 | 480 |
| | Trost et al. (1997) | 0.9 (268.0) | 0.08 | 294.15 ± 0.5 (vapor pressure)[d] | $2.1 \times 10^{-4}$ | 2,000 |
| Para-xylene | This work | 3.19 (272.32) | 0.10 | 298.2 ± 1.0 (225.46 Torr)[b] | 0.01 | 1,000 |
| | Etzkorn et al. (1999) | 2.69 (272.23) | 0.17 | 298.15 ± 3.0 (vapor pressure)[c] | 0.42 | 480 |
| | Trost et al. (1997) | 3.2 (272) | 0.29 | 293.15 ± 0.5 (vapor pressure)[d] | $4.6 \times 10^{-5}$ | 2,000 |

The detection limits, as concentration detected at 1km of light path length, are computed 3 times noise of the spectrometer used.

[a]The differential cross section value after high-pass filtering.

[b]The reported pressures for all gas species are the total pressure (calibrated) with N$_2$ as the gas balance. The
resolutions of this work for ACS measurement of benzene are 0.39 nm; toluene, ortho- and para-xylenes are 0.35 nm; styrene and sulfur dioxide are 0.28 nm.

[c]The resolution reported by Etzkorn et al. (1999) is 0.146 nm.

[d]The resolution reported by Trost et al. (1997) is 0.11 nm.

[e]The resolution reported by Vandaele et al. (1994) is 2 cm$^{-1}$.








**Table 5 Parameters of the NLS fit determined through comparing the convolved spectrum with the PM spectrum. These parameters were then used to optimize the dispersion of pixels for other simulated gas species not measured in the lab.**

| Species | Wavelength region | Non-linear least-squares fit parameters used for ACSs simulation | |
|---|---|---|---|
| | | Squeezing $(a)$ | Shifting $(b)$ |
| Benzene | Region I | 0.989 | $2.22 \times 10^{-14}$ |
| Para-xylene | Region II | 1.000 | -0.033 |
| Sulfur dioxide | Region III | 0.996 | 0.0161 |

Wavelength shifting and squeezing were found by applying optimization process to the convoluted literature

ACS $\sigma_r$ with the reference measured ACS $\sigma_{This\,work}$ using the following relation: $\sigma_{HS} =$

$\kappa xspline(\lambda_r\,(convolution), \sigma_r(convolution), a + bx\lambda_{PM})$. Where a cost function $\chi^2 = \sigma_{PM} - \sigma_{HS}$ is minimized.






**Table 6 Summary of among the ACSs the obtained after applying HS (convolution and nonlinear least-squares fitting) and the measured ACS.**

| Species | Absorption cross section at a strong band near the center wavelength of the instrument function (wavelength) [$10^{-18}$ cm$^2$ (nm)] | | | $\sigma_{HS} - \sigma_{PM}$ Differences of two ACSs [%] (wavelength, %) |
|---|---|---|---|---|
| | $\sigma_{CONV}$ Literature After Convolution [blue] | $\sigma_{PM}$ Lab measurement [red] | $\sigma_{HS}$ Final ACS After HS [black] | |
| Benzene | 2.43 (252.94)[a] | 1.31 (252.96) | 1.32 (252.90) | 0.8 [c] (0.08) |
| Para-xylene | 1.78 (266.59)[a] | 2.39 (266.91) | 2.42 (266.61) | 1.2 [c] (0.11) |
| Sulfur dioxide | 0.96 (302.14)[b] | 1.02 (302.34) | 1.05 (302.31) | 2.9 [c] (0.01) |

[a]The data was from convolution of ACS from (Etzkorn et al., 1999) with IF of the spectrometer

[b]The data was from convolution of ACS from (Vandaele et al., 1994) with IF of the spectrometer

[c]The percentage differences between the HS ACS $\sigma_{HS}$ and the PM ACS $\sigma_{PM}$ were calculated using the relation =

$\frac{|\sigma_{HS} - \sigma_{PM}|}{\sigma_{PM}} x100\%$








**Table 7 Summary of HS ACSs and their detection limits. Simulated by applying the NLS fit parameters to the convolved spectrum. Detection limits were calculated assuming a path length of 1,000 m.**

| Species | Literature | The IF used for convolution | ACS of the strongest band (wavelength) $10^{-18}$ cm$^2$ (nm) and ACS uncertainty (%) | Temperature (Pressure) K (spec. pressure) | Detection limit (ppb) |
|---|---|---|---|---|---|
| Oxygen | Volkamer et al. (1996); Volkamer et al. (1998) | Region I Region II | $-2 \times 10^{18}$ (251.93) and 5.24 $-4.7 \times 10^{18}$ (251.99) and 5.31 | 294 (ATM) | 0.31 0.43 |
| Ozone | Burrows et al. (1999); Voigt et al. (2001) | Region I Region II Region III | 6.86 (251.86) and 2.83 16.45 (251.93) and 2.83 13.76 (251.67) and 3.51 | 293 (ATM) | 0.04 0.19 0.38 |
| Ethylbenzene | Etzkorn et al. (1999) | Region II | 0.94 (259.99) and 4.19 | $298.15 \pm 3.0$ (vapor pressure) | 0.32 |
| Phenol | Volkamer et al. (2001) | Region II | 27.32 (275.13) and 4.13 | 298 (in air) | 0.19 |
| Meta-xylene | Trost et al. (1997) | Region II | 2.20 (263.44) and 0.74 | $293.65 \pm 0.5$ (vapor pressure) | 0.17 |
| Para-cresol | Trost et al. (1997) | Region II | 9.44 (282.76) and 0.75 | $295.15 \pm 0.5$ (vapor pressure) | 0.05 |
| Formaldehyde | Meller et al. (2000) | Region III | 0.13 (326.67) and 0.95 | 298 ($0.5 \pm 0.05$ Torr) | 0.30 |
| Nitrogen dioxide | Stutz et al. (2000) | Region III | 0.60 (355.64) and 1.01 | $323.15 \pm 0.1$ (ATM) | 0.80 |
| Benzaldehyde | Etzkorn et al. (1999) | Region III | 7.14 (284.11) and 0.25 | $298.15 \pm 3.0$ (vapor pressure) | 0.20 |





**Table 8 Uncertainty budget table for HS ACS proposed in this study.**

| Species | Literature ACS uncertainty, $u(\sigma_r)$ | IF Uncertainty Propagation to Convolution, $u(IF_{convol})$ | Fitting Error, $u(Fit)$ | Total Uncertainty, $u(\sigma_{HS}(\lambda))$ |
|---|---|---|---|---|
| Oxygen | 5.0% for both Volkamer et al. (1996) and Volkamer et al. (1998) | 1.58% for Volkamer et al. (1996) and 1.78% for Volkamer et al. (1998) | 0.073% for Volkamer et al. (1996) and 0.12% for Volkamer et al. (1998) | 5.24% for Volkamer et al. (1996) and 5.31% for Volkamer et al. (1998) |
| Ozone | 2.6% for Burrows et al. (1999) and 3.0% for Voigt et al. (2001) | 1.11% for Burrows et al. (1999) and 1.81% for Voigt et al. (2001) | 0.073% for Burrows et al. (1999) and 0.205% for Voigt et al. (2001) | 2.83% for Burrows et al. (1999) and 3.51% for Voigt et al. (2001) |
| Ethylbenzene | 4.12 | 0.73 | 0.12 | 4.19 |
| Phenol | 4.06 | 0.73 | 0.12 | 4.13 |
| Meta-xylene | 0.09 | 0.73 | 0.12 | 0.74 |
| Para-cresol | 0.13 | 0.73 | 0.12 | 0.75 |
| Formaldehyde | 0.05 | 0.93 | 0.205 | 0.95 |
| Nitrogen dioxide | 0.46 | 0.87 | 0.205 | 1.01 |
| Benzaldehyde | 0.05 | 0.14 | 0.205 | 0.25 |







**Table 9 Comparison of concentration obtained from fitting results using the DOAS technique and the gravimetric technique.**

| Species | Fitting value (Standard uncertainty ($\sigma$) [ppm] | CRM gas value (Standard uncertainty ($\sigma$)) [ppm] (Cylinder D068005) | Difference [%] |
|---|---|---|---|
| Benzene | 111.4 (3.2) | 102.6 (0.5) | 8.5 |
| Toluene | 102.6 (3.2) | 101.0 (0.5) | 1.6 |
| Ortho-Xylene | 105.7 (3.4) | 100.8 (0.5) | 4.9 |
