# Peer review of "Preparation of hybrid calibrated absorption cross sections for a compact UV-DOAS measurement"

_Atmospheric Measurement Techniques, 2024_

## Author Comment (AC1)

Reply to the reviewer 1's comment

Before entering the reply section, we express our gratitude to Reviewer 1 for her/his careful and detailed reading of the manuscript and the insightful and helpful comments, which we were pleased to follow up. We believe that addressing all of reviewer's comment should improve a quality of our manuscript. Therefore, we hope to try address all reviewers' comment properly. Please find our reply color-coded as blue between reviewers' comments.

Reviewer #1:

The manuscript provides a detailed discussion of absorption cross-sections using DOAS technology, and the experimental procedures are described comprehensively. However, in its current form, the manuscript requires substantial revision to improve clarity and consistency.

**Major Concerns:**

1. The manuscript's writing and structure are often confusing, making it difficult to understand the main arguments and methods. A notable example is the inconsistent definition of the hybrid simulation (HS) method for absorption cross-sections (ACSs). For instance, in Lines 163–166:

   *Hybrid simulation (HS) of ACSs involves converting the reference absorption spectrum from the literature using the instrument function (IF) to simulate how the absorption spectrum would appear if measured by the specific spectrometer.*

   However, after Section 3.3, the HS method appears to change its definition to include convolution and nonlinear least-squares (NLS) fitting. Throughout the manuscript, the term "HS method" is used multiple times, and it is unclear whether it refers to a standard methodology or a novel approach proposed by the authors, named the hybrid calibration method. More clarity and differentiation are necessary.

   - On the major comments:

   following the recommendation of the three reviewers who found it difficult to follow the flow of this manuscript, we agreed to split the paper into two parts. The first part is about the hybrid simulation (HS) method and the second part is about the precise measurement (PM) method.

   *Our proposed Hybrid Simulation (HS) method combines two key steps: first, convolving the reference spectrum with the instrument function (IF) of the spectrometer, and second, applying nonlinear least-squares (NLS) fitting.*

   This definition of the HS method is consistently applied throughout the revised manuscript to avoid any confusion.

2. Similarly, there is ambiguity regarding the blue line in Figure 5—does it represent the high-resolution cross-section or the convoluted cross-section? The manuscript appears to present conflicting descriptions, which require resolution for accurate interpretation.

   - I apologize for the confusion due to the ambiguity in the parameters in the manuscript. We have fixed this problem by making a consistent parameter throughout the new manuscript as follows:

   $\sigma(\lambda)_{meas}$ : Absorption cross section spectrum measured at wavelength $\lambda$

   $\sigma(\lambda)_{HS}$ : Absorption cross section spectrum simulated using HS approach at wavelength $\lambda$

   $\sigma(\lambda)_{ref}$ : Reference absorption spectrum from literature

   The blue line in Figure 5 (now become Figure 3 in the new manuscript) is refers to the reference ACS from literature $\sigma(\lambda)_{ref}$. All others parameters are also corrected accordingly in the graph and the main text.

3.  As stated by authors, commonly used cross-sections are either convoluted or experimentally measured. My understanding is that the authors have proposed an innovative method that not only convolutes the high-resolution cross-sections but also applies NLS fitting. This method could potentially be extended to other gases within the same spectral range where experimental cross-sections are unavailable (e.g., Line 295). While this seems a promising innovation, I have reservations about its broader applicability. Absorption cross-sections reported by different studies often exhibit inherent differences, raising doubts about the validity of applying the same correction factor across various studies. To address these concerns, I suggest that the authors provide additional experimental evidence, such as simultaneous measurements of benzene and toluene cross-sections within the same spectral range, comparing the correction factors obtained through separate NLS fittings.

-A comparison experiment between the correction factors obtained using the HS method for benzene and toluene has been performed for the measurement range between 257.9 and 269.6 nm. For the comparison purpose, we made a similar treatment to the both spectra. In this experiment, we use IF at 265 nm for benzene and toluene. However, in the manuscript we used IF at the center wavelength of 248 nm for benzene because in the real application of LP-DOAS we focus on the deep UV range where less strong absorption of other gas species is observed in the range near 248 nm.

**Benzene**

[Figure]

**Figure 1**. Comparison of the laboratory-measured ACS, the reference literature ACS, and the ACS generated using the HS approach. The convolution alone cannot align the reference ACS spectrum with the measured ACS spectrum. However, applying the HS method improves the fit, as indicated by a reduction in the Sum of Squared Errors (SSE), from $1.6004 \times 10^{-36}$ before NLS fitting to $4.2574 \times 10^{-37}$ after NLS fitting.

**Table 1**. The optimized parameters from NLS fitting from benzene experiment.

| Parameter | Value |
|---|---|
| $a$ (squeezing) | 1.001 pixel (0.0635 nm) |
| $b$ (shifting) | -1 pixel (-0.063 nm) |
| $c$ (amplitude scaling) | 0.6 cm$^2$·molecule$^{-1}$ |

**Toluene**

[Figure]

**Figure 2**. Comparison of the laboratory-measured ACS, the reference literature ACS, and the ACS generated using the HS approach. The convolution alone cannot align the reference ACS spectrum with the measured ACS spectrum. However, applying the HS method improves the fit, as indicated by a reduction in the Sum of Squared Errors (SSE), from $4.5 \times 10^{-36}$ to $1.45 \times 10^{-36}$.

**Table 2**. The optimized parameters from NLS fitting from toluene experiment.

| Parameter | Value |
|---|---|
| $a$ (squeezing) | 1.003 pixel (0.0636 nm) |
| $b$ (shifting) | -5 pixel (-0.317 nm) |
| $c$ (amplitude scaling) | 0.68 cm$^2$·molecule$^{-1}$ |

We compared the fitting parameters between the two gas species and observed good agreement between the amplitude scaling and squeezing parameters for the HS method. As shown in Figure 1 and Figure 2, when relying solely on convolution, the peak intensity of the convolved reference spectrum does not align well with that of the measured spectrum. However, by applying the HS procedure, the peak intensities of the simulated spectra more closely match the measured spectra. This improvement is reflected in reduced sum of squared errors (SSE) values for both the benzene and toluene measurement spectra. This demonstrates that our approach, which integrates both convolution and nonlinear least squares (NLS) fitting, enhances the accuracy of spectral simulation and is applicable to other gases within the same spectral range where experimental cross-sections are unavailable. Although there is a slight discrepancy of approximately 0.25 nm in the shifting coefficient between the two different literature absorption cross-sections, which may be attributed to variations in experimental conditions, the key parameters—amplitude and squeezing—remain consistent between the two HS simulation results at a same spectral range. The squeezing parameter represents spectral broadening, while the amplitude indicates the depth of absorption (i.e., the optical density of the gas absorber). The shifting parameter, on the other hand, can be effectively determined and corrected using NLS fitting during LP-DOAS retrieval.

4. The correction factors presented in Table 5 are extremely small. Would these corrections not already be manageable within the DOAS fitting process without requiring the proposed methodology? Additional clarification is needed to justify the necessity of this approach.

-In response to the comment regarding the small correction factors in Table 5, we would like to clarify that the amplitude scaling parameter ($c$), which is critical as it indicates the depth of absorption, was not included in the Table. This parameter represents the optical density of the gas absorber and cannot be fully accounted for through the standard DOAS fitting process alone. The DOAS fitting primarily corrects for absorption features, but it does not directly adjust for the amplitude scaling for the reference spectrum. Therefore, we added this parameter in Table 5 serves to emphasize its importance and to clarify why it is necessary to incorporate our proposed methodology for a more accurate spectral analysis.

**Minor Concerns:**

1. Figure 5 appears before Figure 3 in the manuscript, which disrupts the logical flow. Please reorder the figures appropriately.

-Thank you for your comments. We now divide the paper into two parts: hybrid simulation (HS) and precise measurement (PM) to make it easier for the reader to follow. Figure 3 will be discussed in the PM part and has been deleted from the first part of the paper.

2. Line 303: The subscripts for ($O_2$) and ($O_3$) are incorrect and require revision.

-We have fixed it as suggested.

3. Table 4: The detection limits reported for Trost et al. seem unusually low—three orders of magnitude smaller than other devices. Please verify these values.

-The reason why DL of Trost et al much smaller than other is because the assumed path length is longer than the other work. We will discuss it more detail in the second part of the manuscript. We have deleted the table in the first part of the manuscript because it is related to the PM approach.

4. Table 5: The drift for benzene is reported as ($10^{-14}$). Such an exceedingly small number raises questions about its significance—please clarify.

-The value is accurate because we have calibrated our spectrometer's wavelength, ensuring precise readings that closely match the literature values. The main difference lies in the amplitude scaling (c) parameter, which we have now included in the new draft.

-------------------------------------------------End of the document-------------------------------------------------

---

## Author Comment (AC2)

**Reply to the reviewer 3's comment**

Before entering the reply section, we express our gratitude to Reviewer 3 for her/his careful and detailed reading of the manuscript and the insightful and helpful comments, which we were pleased to follow up. We believe that addressing all of reviewer's comment should improve a quality of our manuscript. Therefore, we hope to try address all reviewers' comment properly. Please find our reply color-coded as blue between reviewers' comments.

Reviewer #3:

The manuscript of Trisna and co-authors applies the well-established active DOAS (Differential Optical Absorption Spectroscopy) technique in the deep UV spectral range between 230 and 320nm. The authors set up an own custom build active DOAS system for the measurements of aromatic compounds like benzaldehyde, formaldehyde, NO2, p-cressol, m-xylene, phenol, ozone and other in a measurement cell. The instrument setup was very basic with a low-quality compact spectrometer and without fulfilling quality criteria for such DOAS instruments like described in Platt and Stutz (2008).

The authors observed a difference between modeled absorption spectra and measured absorption spectra of the system with reference gases. The manuscript describes the development of a hybrid calibration of the UV DOAS system combining modeled and measured absorption spectra. From the topic this fit in the scope of AMT. While the development of this hybrid calibration seems to be reasonable and may be scientifically valuable, I cannot recommend this manuscript for publication due to the following severe issues:

1. The measurement setup does not fulfil basic quality criteria for such active DOAS systems (details on the obvious quality issues listed below). With a poor setup any measurement technique likely derives wrong measurement results. It is no surprise that a high difference between modeled and measured SO2 absorption spectra of up to 62% (p. 4 l. 93) is observed. This is not due to the DOAS measurement technique but due to the setup and data processing. The authors did not try to improve the instrument and correct instrument effects. This is a lack of thoroughly scientific work. I would expect that the differences between modeled and measured absorption cross section will almost disappear with appropriate setup and data correction.

   -Thank you for your comments. While we did not go into detail in the original manuscript, we would like to clarify that we have indeed taken steps to correct for noise in the spectrometer, particularly addressing the electronic offset (EO) and dark current (DC). Additionally, we maintained the detector temperature at the lowest setting (278.15 K) during operation to minimize noise. We also performed wavelength calibration using the Hg emission lamp, ensuring that the spectrometer's wavelength accuracy was properly established. With these corrections in place, we are confident that the data we present meets high scientific standards.

   To prevent any confusion, we will include more detailed information on the spectrometer's characterization in the revised manuscript.

   Regarding the discrepancy between the simulation results with convolution and the measured data, we would like to note that such differences are not unique to our study. Similar discrepancies have been observed in other studies, including those using more precise instruments like FTIR. For example, Fally (2009) reports similar deviations when comparing FTIR measurements with literature values (Figure 2, Figure 4 and Equation (4) in their paper).

2. The setup feature basic lack of instrument requirements:

   a. The use of a xenon arc lamp (p. 5 l. 117) will cause a high rate of unused light which will act as stray light, as the main emission range of this lamp is in the range of 350 to 600 nm. This will cause a high stray-light amount. No reduction and characterization of stray light is applied which will be relevant for these measurements. Measurements in this spectral range would better use deuterium lamps or LEDs.

   -We chose a xenon lamp as the light source, despite its high stray light, because we aimed to apply

this system for field measurements. The high intensity provided by the xenon lamp is essential for effective measurements at distances of up to 1.6 km. To mitigate stray light, we use solarization-resistant fiber optics, which help attenuate unwanted light. Additionally, xenon lamps are commonly used in deep UV measurements of ACSs and DOAS applications by other research groups (Lampel, J., 2018; Trost., 1997; Wang, M., 2024)

b.  The used spectrometer AVASpec-ULS2048LTEC (p.5 l. 131) is not well suited for this spectral range. It features a very low quantum efficiency of ~ 10% in the UV spectral range, causing not only a low signal, but also a high stray light. A spectrometer with a back-thinned or CMOS detector are much better suited for this deep UV spectral range due to a higher quantum efficiency.

-We have thoroughly characterized the noise and stray light in the detector and found that, under optimal conditions, the detector performs well. By maintaining the detector temperature at 278.5 K, setting the exposure time to 0.02 s, and averaging 100 spectra, we were able to keep the noise below 1000 counts and achieve SNRs of 4442 for benzene, 4253 for p-xylene, and 383 for $SO_2$. These results are more than adequate for field measurements in LP-DOAS applications. For more detailed information, please refer to Section 2.3.1 in the new draft.

c.  Stray light in the spectrometer is not characterized and corrected. The system will have significant stray light which requires correction.

-Although we did not explicitly mention it in the original paper, we have characterized and corrected both the dark current (DC) and electronic offset (EO). This process is now discussed in detail in the updated version of the manuscript in Section 2.3.1. While the spectrometer may exhibit some stray light, we have implemented appropriate correction measures to minimize its impact on the measurements.

d.  The spectrometer configuration with an entrance slit of 50 μm width (p. 5 l. 133) would cause with this spectrometer a spectral under-sampling of the instrument line function. The full width half maximum should be minimum 5 to 6 pixels (Platt and Stutz, 2008), but here it will be ~ 4 pixel. Modeling absorption spectra can thus be wrong.

-The 50 μm slit size does not lead to spectral under-sampling in our spectrometer, as we carefully optimize both the exposure time and spectral averaging to achieve a high SNR, reaching up to 4442 for benzene. Additionally, a 50 μm or even lower slit size is commonly used in deep UV studies and has been shown to produce reliable spectra (Retzer, U., (2020); Wang, M., (2024)).

e.  It looks like non-linearity of the spectrometer detector are not considered. The applied detector has a very strong non-linear response influencing the measurements.

-We have thoroughly considered the issue of non-linearity in the spectrometer, particularly in the wavelength calibration and NLS fitting procedures. To address the nonlinearity in the pixel-wavelength mapping, we used a quadratic equation during wavelength calibration. As shown in Figure 4(c), the quadratic approach significantly reduces the wavelength deviation from actual value compared to a linear fit. Additionally, if any residual non-linearity remains in the detector's response, the NLS fitting process accounts for it by incorporating adjustments such as wavelength shifting, squeezing, and amplitude scaling. This ensures that the non-linearity is effectively addressed.

f.  The instrument line function of the spectrometer (Figure 2) is not acceptable. Compact spectrometers have a limited quality, but this is much worse than normal. Avantes AVASpec-ULS spectrometers have a much better instrument line function similar to Gaussian shape which do not vary much over the spectral range (when correctly adjusted). Also, the baseline around the instrument line function is not zero and varying over the spectral range. This may indicate a stray light issue or defect spectrometer grating. Something is clearly wrong with this spectrometer or setup.

-Figure 2 shows the raw data from the spectrometer, prior to any EO (electrical offset) and DC (dark current) corrections. However, during the convolution process, we first correct for both EO and DC to ensure the baseline is at zero. We will change the Figure with the EO and DC corrected one.

-Regarding the instrument line function (IF), it is true that in spectrometers with a large number of pixels and a short wavelength range, the IF remains relatively constant. In our setup, however, the broad measurement range (237 nm to 357 nm) combined with 2048 pixels naturally leads to variations in the IF due to dispersion effects (Palmer, C., (2005)). We choose this spectrometer because it offers broad dynamic range so that many gas species can be detected simultaneously.

g. The quality of the wavelength calibration is not described which may have here a significant contribution to the difference.

-We have performed wavelength calibration on the spectrometer, which is why the wavelength shifting observed in the NLS fitting results is relatively small. Without this calibration, the shifting value would likely be much larger. A detailed discussion of the wavelength calibration process has been added in the latest draft, specifically in Section 2.3.2. Please refer to this section for further information.

3. The measurement with calibration bottles ignores losses and reaction of the gas in the bottle, on cell walls, inlets etc.. The gas concentration in the measurement cell may be different than expected. This is a big disadvantage of the reference gas method in comparison to the modeled absorption cross section method and can have a significant influence on the comparison.

- The modelled ACS has inherent limitations, as it is based on idealized conditions that may not fully represent the real-world operating conditions of the measurement system. Therefore, the goal of this study is to quantify the difference between the modelled and measured ACS and refine the modelled spectrum to improve its accuracy.

-While we acknowledge that using gas in a bottle for calibration has some inherent challenges, such as potential losses or reactions of the gas with the bottle walls, inlets, or other components, it remains one of the most accurate preparation methods. This approach combines gravimetric techniques with detailed analysis of each uncertainty component, including transfer between vessels or cylinders (Milton et al., 2011). Compared to syringe injection techniques, the gravimetric method offers significantly higher accuracy, with an uncertainty of less than 1.3% (Grenfell et al., 2010). Although there may be slight discrepancies in gas concentration due to system imperfections, these are typically well-controlled and do not undermine the overall reliability of the reference gas method.

**Minor technical points:**

p. 4 l. 88: The convolution not necessarily need to assume a constant instrument line function. A broadening can be simulated and included in the modeling.

-We added a sentence: "*It is also important to note that convolution does not necessarily require a constant IF, as broadening effects can be simulated and incorporated into the modeling.*" to point out this comment.

p. 4 l. 101: The temperature dependence of ACSs are in IR large, but almost ignore-able in the UV.

-We added a sentence: "*While the temperature dependence of ACSs is significant in the IR range, it is nearly negligible in the UV region.*" to add this point.

p. 5 l. 133: What grating type? What is the blaze wavelength?

-The grating type is blazed grating and the blaze wavelength is in the UV range. The order code UE in this manual: https://web.unideb.hu/uh9v32/muszer2/avantes_all.pdf.

p. 10 l. 283: The DOAS fit procedure can also include shift and squeeze of the modeled ACS. This is here required due to an insufficient wavelength calibration accuracy. A correction for the hybrid simulation is understandable, but it is not a general issue for the DOAS fit procedure.

-We added this sentence to clarify the important of HS procedure in the preparation of ACS database

for DOAS fitting *"In the DOAS fitting procedure, similar parameters, including shift and squeeze, can be incorporated if needed. However, for the HS measurements, the amplitude scaling parameter is particularly important, as it directly influences the gas concentration value and is strongly correlated with the optical depth of the reference spectrum."*

Figure 1: What is the fiber optic type and gas cell type?

-The fiber optic cable is a UV-VIS XSR solarization-resistant model from Ocean Optics, and the gas cell is a quartz glass hollow tube. We added this detail in the revised manuscript.

Figure 4: The measured and modeled ACSs (upper panel) seem to agree better than the comparison between the measured and the hybrid simulated ACSs (lower panel). So hybrid is worse than modeled? This seem to be wrong.

-We apologize for the confusion. Figure 4 does not show a comparison between ACS from the model and ACS from the hybrid simulation (HS). Instead, Figure 4 presents ACS values obtained from laboratory measurements, with the black line representing the mean value and the blue and red lines indicating the measurement uncertainties. We confirm that the measurements within the shaded gray area (between the red and blue lines) are accurate.

Figure 5: The large difference of the modeled ACSs depending of the instrument line function indicate a spectrometer or setup issue. They should be very similar.

-This behavior is typical for grating-type spectrometers, especially in setups with a wide dynamic range. Due to the limited number of pixels, the spectrometer's resolution is optimal only at the blaze wavelength. As a result, intensity variations can occur depending on the instrument's line function (IF), leading to discrepancies in the detected ACSs.

The authors need first to quantify if the differences are due to the low-quality setup and what cause the problems. Second the effects like stray light need to be corrected as good as possible. Afterwards I can see a scientific relevance for the manuscript if the hybrid calibration is developed as a tool for low quality UV-DOAS systems (which do not fulfill typical DOAS quality criteria) to allow such systems still to derive representative measurement data. However, the manuscript than need a complete rewriting as the focus will change. A misunderstanding that the correction is needed due to the DOAS technique or due to the use of compact spectrometers must be avoided.

-As we have discussed, the discrepancy between the modelled and laboratory-measured ACS when relying solely on convolution has not only been observed in our study but also in other research using higher-resolution instruments, such as FTIR, as noted by Fally (2009). In response, Fally developed the NLS method to address this issue. In our study, we applied a similar approach to a grating-type spectrometer, specifically to overcome the non-linearity of the spectrometer's readings. This discrepancy is not due to the low quality of the setup, but rather an inherent challenge in the convolution method when applied to spectrometers with lower resolution.

To clarify, we have already quantified the potential sources of the discrepancies, including stray light and other detector-related issues, and have implemented corrections to mitigate these effects as thoroughly as possible. The hybrid calibration method we developed is intended to enhance the performance of lower resolution UV-DOAS systems for field application purpose.

Reference:

Fally, Sophie, Michel Carleer, and Ann C. Vandaele. "UV Fourier transform absorption cross sections of benzene, toluene, meta-, ortho-, and para-xylene." Journal of Quantitative Spectroscopy and Radiative Transfer 110.9-10 (2009): 766-782.

Grenfell, R. J. P., Milton, M. J. T., Harling, A. M., Vargha, G. M., Brookes, C., Quincey, P. G., & Woods, P. T. (2010). Standard mixtures of ambient volatile organic compounds in synthetic and whole air with stable reference values. Journal of Geophysical Research: Atmospheres, 115(D14).

Lampel, J., Zielcke, J., Schmitt, S., Pöhler, D., Frieß, U., Platt, U., & Wagner, T. (2018). Detection of O 4 absorption around 328 and 419 nm in measured atmospheric absorption spectra. Atmospheric Chemistry and Physics, 18(3), 1671-1683.

Milton, M. J. T., Vargha, G. M., & Brown, A. S. (2011). Gravimetric methods for the preparation of standard gas mixtures. Metrologia, 48(5), R1.

Palmer, C. and Loewen, E. G.: Diffraction grating handbook, 2005.

Retzer, U., Ulrich, H., Bauer, F. J., Will, S., & Zigan, L. (2020). UV absorption cross sections of vaporized 1-methylnaphthalene at elevated temperatures. Applied Physics B, 126, 1-6.

Trost, B., Stutz, J., & Platt, U. (1997). UV-absorption cross sections of a series of monocyclic aromatic compounds. *Atmospheric Environment*, *31*(23), 3999-4008.

Wang, M., Connolly, S. C., & Venables, D. S. (2024). Deep-ultraviolet absorption cross sections of strongly absorbing atmospheric species. Journal of Quantitative Spectroscopy and Radiative Transfer, 323, 109050.

-------------------------------------------------End of the document-------------------------------------------------

---

## Author Comment (AC3)

**Reply to the reviewer 2's comment**

Before entering the reply section, we express our gratitude to Reviewer 2 for her/his careful and detailed reading of the manuscript and the insightful and helpful comments, which we were pleased to follow up. We believe that addressing all of reviewer's comment should improve a quality of our manuscript. Therefore, we hope to try address all reviewers' comment properly. Please find our reply color-coded as blue between reviewers' comments.

Reviewer #2:

The manuscript by Trisna and co-authors employs the widely-used active DOAS (Differential Optical Absorption Spectroscopy) technique in the deep UV range of 230–320 nm. The authors developed a custom-built active DOAS system to measure aromatic compounds, including benzaldehyde, formaldehyde, $NO_2$, p-cresol, m-xylene, phenol, ozone, and others within a measurement cell. The instrument setup was relatively simple, utilizing a low-quality compact spectrometer.

As Anonymous Referee #3 pointed out, since literature absorption cross-sections are typically measured with various types of spectrometers, it is essential to more thoroughly discuss the properties of the spectrometer used in this study.

To begin, I strongly encourage you to incorporate the valuable feedback from both Anonymous Referee #3 and Anonymous Referee #1. Furthermore, I would like to remind you of a few suggestions from the access review process that have only partially been addressed:

**Consider splitting the manuscript into two separate papers:** One focusing on the method for applying both measured and simulated literature absorption cross-sections, and another dedicated to the actual measured absorption cross-sections. The current manuscript lacks clarity regarding the method combining measured and convolute cross-sections, which is challenging to follow.

A separate analysis of the measured absorption cross-sections could be highly valuable for the scientific community, particularly if it includes more detailed comparisons with existing literature data, where not a lot of published laboratory measurements are currently available. For example, see the plots provided in Serdyuchenko et al. (2014).

- On the major comments:

Following the recommendation of the three reviewers who found it difficult to follow the flow of this manuscript, we agreed to split the paper into two parts. The first part is about the hybrid simulation (HS) method and the second part is about the precise measurement (PM) method. We will submit the second part separately.

**Further investigate the instrument function on the technical side:** This type of instrument generally produces higher-quality spectra. Ensure the slit and/or fibers are well and uniformly illuminated. Additionally, if using a xenon arc lamp in this wavelength range, the instrument may experience notable impacts from stray light within the spectrometer. These effects should be measured and discussed. Refer to comments from Anonymous Referee #3 for additional insights.

-Regarding the quality of the spectra, although it is not discussed in detail in the initial draft, we have already performed wavelength calibration, electronic offset (EO), and dark-current (DC) corrections on the ACS measurement data we provided. This ensures the reliability and high quality of the data.

-The discrepancy between laboratory measurements and simulations, when using only the convolution procedure, has also been observed in other studies, such as those by Fally et al (2009), even when more accurate instruments like FTIR were used (see Figure 2, Figure 4, and Equation 4 in their published manuscript).

-To prevent any misunderstandings, we will include additional details on wavelength calibration, electronic offset (EO), and dark-current (DC) noises corrections in the paper.

-We have thoroughly characterized the noise and stray light in the detector and found that, under optimal conditions, the detector performs well. By maintaining the detector temperature at 278.5 K, setting the exposure time to 0.02 s, and averaging 100 spectra, we were able to keep the noise below 1000 counts and achieve SNRs of 4442 for benzene, 4253 for p-xylene, and 383 for $SO_2$. These results are more than adequate for field measurements in LP-DOAS applications. For more detailed information, please refer to Section 2.3.1 in the new draft.

**The data analysis and explanation of processes and variables** remain somewhat challenging to follow. Thank you for adding a list of acronyms following the access review, which has improved clarity (a bit).

-We apologize for any confusion regarding the data analysis and explanation of the variables. We have fixed this issue by carefully proof read the draft and fix the variables and parameters. Here are the main parameters that often giving confusion:

$\sigma(\lambda)_{meas}$ : Absorption cross section spectrum measured at wavelength $\lambda$

$\sigma(\lambda)_{HS}$ : Absorption cross section spectrum simulated using HS approach at wavelength $\lambda$

$\sigma(\lambda)_{ref}$ : Reference absorption spectrum from literature

Lastly, please address the issue in **Figure 6**: Applying a convolution should not alter the absorption cross-section itself, as the broadband contribution should be uniform across all instrument line functions. This discrepancy requires further discussion, if not revision. If this data represents fitted concentrations, it does not reflect the absolute absorption cross-section and may be misleading.

-You are correct that convolution itself cannot alter the absorption cross-section (ACS), as the broadband contribution should be uniform across all instrument line functions. The discrepancies observed in Figure 6 arise from the NLS fitting process, specifically from the multiplication parameter ($c$). Due to the spectrometer's uneven spectral resolution, the NLS fit generates different multiplication parameters depending on the fitting region. This variation is a result of the specific spectral range used for the fitting process.

-To address this, we have revised Figure 6 (now Figure 7 in the new draft) by replacing the "IF" with the region number, which better reflects the fitting regions and clarifies the results. We believe this adjustment resolves the discrepancy and provides a clearer understanding of the data.

Reference:

Fally, Sophie, Michel Carleer, and Ann C. Vandaele. "UV Fourier transform absorption cross sections of benzene, toluene, meta-, ortho-, and para-xylene." Journal of Quantitative Spectroscopy and Radiative Transfer 110.9-10 (2009): 766-782.

--------------------------------------------------------End of the document--------------------------------------------------------